# Agreement and Reproducibility of Anterior Chamber Angle Measurements between CASIA2 Built-In Software and Human Graders

**DOI:** 10.3390/jcm12196381

**Published:** 2023-10-06

**Authors:** Gustavo Espinoza, Katheriene Iglesias, Juan C. Parra, Ignacio Rodriguez-Una, Sergio Serrano-Gomez, Angelica M. Prada, Virgilio Galvis

**Affiliations:** 1Centro Oftalmológico Virgilio Galvis, Floridablanca 681004, Santander, Colombia; 2Fundación Oftalmológica de Santander, Floridablanca 681004, Santander, Colombia; 3Facultad de Ciencias de la Salud, Universidad Autónoma de Bucaramanga, Bucaramanga 680002, Santander, Colombia; 4Instituto Universitario Fernández-Vega, Fundación de Investigación Oftalmológica, Universidad de Oviedo, 33012 Oviedo, Spain; irodriguezu@fernandez-vega.com

**Keywords:** agreement, anterior chamber angle, AS-OCT, optical coherence tomography, primary angle closure glaucoma, reproducibility

## Abstract

Purpose: This study evaluated the agreement and reproducibility of ACA measurements obtained using the built-in software of the CASIA2 (Version 3G.1) and the measurements derived from expert clinicians. Methods: Healthy volunteers underwent ophthalmological evaluation and AS-OCT examination. ACA measurements derived from automated and manual SS location were obtained using the CASIA2 automated software and clinician identification, respectively. The intraobserver, interobserver reproducibility, CASIA2–human grader reproducibility and CASIA2 repeatability were assessed using intraclass correlation coefficients (ICCs). Results: The study examined 58 eyes of 30 participants. The CASIA2 software showed excellent repeatability for all ACA parameters (ICC > 0.84). Intraobserver, interobserver, and CASIA2–human grader reproducibility were also excellent (ICC > 0.87). Interobserver agreement was high, except for nasal TISA500, differing between observers 1 and 2 (*p* < 0.05). The agreement between CASIA2 measurements and human graders was high, except for nasal TISA500, where observer 1 values were smaller (*p* < 0.05). Conclusion: The CASIA2 built-in software reliably measures ACA parameters in healthy individuals, demonstrating high consistency. Although a small difference was observed in nasal TISA500 measurements, interobserver and CASIA2–human grader reproducibility remained excellent. Automated SS detection has the potential to facilitate evaluation and monitoring of primary angle closure disease.

## 1. Introduction

Primary angle closure glaucoma (PACG) is a major cause of glaucoma-related blindness globally [1]. Angle closure is characterized by appositional approximation or contact between the iris and trabecular meshwork.

Visualization of the anterior chamber angle (ACA) is crucial for diagnosing glaucoma, especially angle closure variants. Although gonioscopy is the gold standard, it has limitations such as subjectivity, dependence on clinician expertise and patient cooperation. In fact, previous studies conducted in the United States showed that many ophthalmologists do not include gonioscopy in their routine examination or even in the examination of glaucoma patients [2,3].

New anterior segment imaging tools, such as anterior segment optical coherence tomography (AS-OCT), offer objective and quantitative ACA analysis [4]. The CASIA2 (Tomey Corporation, Nagoya, Japan) is the second generation of swept-source AS-OCT with a wavelength of 1310 nm and an acquisition speed of 50,000 A-scans/s which allows to perform cross-sections of the anterior chamber 16 mm wide.

The trabecular iris space area (TISA) and angle opening distance (AOD) are some of the measurements used to evaluate the dimensions of the ACA. Previous studies have confirmed a strong relationship between these measurements and gonioscopic angle closure [5,6,7]. The accuracy of these parameters depends on the location of the scleral spur (SS). The CASIA2 incorporates the possibility of automatic detection of the SS. However, the location of the SS is usually edited by a human grader to achieve a more precise location. Therefore, we consider it important to assess the accuracy of the built-in software to adequately measure these parameters and compare it with the measurements derived from the semi-automated SS location.

The aim of this study was to evaluate the agreement between the ACA measurements obtained by the built-in software of the CASIA2 and the measurements derived from expert clinicians. Furthermore, we evaluated the intraobserver, interobserver reproducibility, CASIA2–human grader reproducibility, and CASIA2 built-in software repeatability of these measurements.

## 2. Materials and Methods

This study included healthy volunteers evaluated at the Fundación Oftalmológica de Santander “Carlos Ardila Lülle” (FOSCAL), Colombia. This study was approved by the ethics committee board of Fundación Oftalmológica de Santander (reference: 002338) and adhered to the tenets of the Declaration of Helsinki. Informed consent was obtained from each participant.

### 2.1. Participants

The inclusion criteria were age ≥18 years, normal ophthalmological evaluation, intraocular pressure <21 mmHg, open angles confirmed by gonioscopy with the Shaffer classification ≥3, best-corrected visual acuity using the Snellen chart ≥20/40, refractive error between +3 diopters and −6 diopters of spherical equivalent and cylinder ≤3 diopters, normal visual field with a glaucoma hemifield test, and mean deviation within normal limits. The exclusion criteria were a history of laser procedures, intraocular surgery, use of topical medications that could modify the pupil size, unreliable visual field tests (false-negative and/or false-positive rates ≥33% and/or fixation losses ≥20%), and poor quality AS-OCT images.

All participants underwent ophthalmological evaluation by a single evaluator (G.E.) that included refraction, slit-lamp biomicroscopy, Goldmann applanation tonometry, gonioscopy under low-light conditions, and fundoscopy. The computerized visual field examination was performed using the Humphrey Field Analyzer^®^ (Carl Zeiss Meditec, Dublin, CA, USA) under the SITA Standard strategy using a Goldmann III stimulus. Retinal nerve fiber layer OCT was also performed using a swept-source OCT device (DRI OCT Triton; Topcon, Tokyo, Japan) after pharmacological pupil dilation.

### 2.2. Sample Size Calculation

A sample size of 30 individuals was determined to observe a correlation coefficient of 0.6, with a type I error of 5%, type II error of 20%, and a 20% rate of data loss (Muestreo, Epidat Version 4.2).

### 2.3. AS-OCT Image Acquisition

After confirming that the ophthalmological evaluation was normal, the AS-OCT examination was performed at least 24 h after the initial assessment to avoid changes in the ACA that could induce modifications in the cornea during clinical examination. Each subject underwent examination of both eyes. After a 5 min adaptation period to standardized low-light conditions at 0 cd/m^2^ (measured with a luxmeter), two AS-OCT captures of adequate quality were obtained for each eye by a single operator. During the examination, subjects were instructed to fixate their gaze on the internal fixation system of the equipment. The upper eyelid was gently raised, and a gentle traction of the lower eyelid was made by the operator taking care to avoid inadvertent pressure on the eyeball. The sulcus 0 + 90 measurement protocol was used for each capture. The image across the horizontal meridian was used for the analysis.

### 2.4. AS-OCT Image Analysis

Initially, a masked observer recorded the data of the nasal and temporal TISA500, TISA750, AOD500, and AOD750 parameters, derived from the automatic identification of SS performed with the CASIA2. The definition of AOD is the distance between the cornea and iris along a line perpendicular to the cornea at a specified distance (in μm) from the SS [8]. TISA, which can be tested and evaluated as well as AOD at several distances from the SS, is defined as the trapezoidal area with the following boundaries: anteriorly, the AOD; posteriorly, the line perpendicular to the plane of the inner corneoscleral wall drawn from the scleral spur to the opposing iris; superiorly, the inner corneoscleral wall; and inferiorly, the anterior iris surface [9]. The CASIA2 software utilizes a basic edge detection technique to identify the location of the scleral spur–uvea edge line, enabling SS detection. After obtaining the CASIA2 automatic ACA measurements, the masked observer intentionally relocated the SS to an incorrect position and saved the scan to minimize observer interpretation biases (Figure 1). Subsequently, the manual localization of the SS was carried out using the CASIA2 software tools (Version 3G.1) to obtain the measurements of these same parameters. The ciliary muscle and the bump method were used to identify the SS (Figure 2) [10]. This evaluation was performed by one glaucoma specialist with experience analyzing AS-OCT (G.E.) and one trained glaucoma fellow (K.I.). The evaluators were masked to the origin of the image data to ensure the anonymity of the eyes and participants. Each examiner evaluated each group of images independently (116 images/examiner). The second evaluation of the set of images per eye was performed 2 weeks after the first analysis.

The repeatability of the ACA parameters of the automated software was evaluated by comparing the data from capture 1 and capture 2 of each eye. For intraobserver reproducibility, evaluation 1 of capture 1 and 2 was compared with evaluation 2 of the same captures. For interobserver agreement analysis, evaluation 1 of capture 1 and 2 of G.E. was compared to evaluation 1 of capture 1 and 2 of K.I. Finally, to compare the agreement between each observer and the CASIA2 automatic parameters, evaluation 1 of capture 1 and 2 was compared to the readings of capture 1 and 2 obtained from the CASIA2 automated software.

### 2.5. Statistical Analysis

A univariate analysis was conducted to examine the sociodemographic and ophthalmological variables. Quantitative variables were analyzed using measures of central tendency and dispersion, such as mean and standard deviation. Qualitative variables were assessed using absolute and relative frequencies.

For the bivariate analysis, a comparison of means for the ACA parameters was performed among the different captures, evaluations, and evaluators. Additionally, an absolute intraclass correlation coefficient (ICC) was calculated to assess the reproducibility among the observers, both within and between different AS-OCT captures. Additionally, CASIA2–human grader agreement of measurements was assessed with an ICC and Bland–Altman plots with mean difference and limits of agreement (LoA).

A significance level (alpha) of 0.05 was employed for all analyses, and the statistical software Stata 16.0 was used to conduct the calculations. It is important to note that an ICC value of less than 0.4 indicates poor reproducibility, while a value between 0.4 and 0.75 suggests fair to good reproducibility. A value exceeding 0.75 indicates excellent reproducibility.

## 3. Results

A total of 60 eyes of 30 healthy volunteers were evaluated. Furthermore, 2 eyes of 2 participants were excluded due to missing images, leaving a total of 58 eyes in the final analysis. Among the 30 subjects, 16 (53.3%) were women and 14 (46.7%) were male. The mean age was 30.4 ± 9.05 years (range 21–64 years).

### 3.1. Repeatability of CASIA2 Automated Software

There were no significant differences (*p* > 0.68) between mean ACA measurements of capture 1 and 2 of the nasal and the temporal TISA500, TISA750, AOD500, and AOD750 (Table 1). Furthermore, repeatability was excellent for each of the parameters evaluated (ICC > 0.84).

### 3.2. Intraobserver ACA Measurements’ Reproducibility

There were no significant differences of mean ACA measurements of the nasal or temporal parameters between the first and the second evaluation of images for both evaluators (range *p* = 0.38 to *p* = 0.71). There was also excellent intraobserver reproducibility of all parameters for both evaluators. The ICCs ranged from 0.96 to 0.98 for observer 1 (G.E.) and from 0.97 to 0.98 for observer 2 (K.I.) (Table 2).

### 3.3. Interobserver ACA Measurements’ Agreement

The nasal TISA500 was smaller for observer 1 compared to the measurement of observer 2 with a mean of 0.161 ± 0.067 mm^2^ and 0.179 ± 0.057 mm^2^, respectively (*p* < 0.05). For all the other ACA measurements, there were no significant differences between the two observers (range *p* = 0.06 to 0.40).

Interobserver reproducibility was excellent for all parameters evaluated. ICC values ranged from 0.90 (nasal TISA500) to 0.96 (temporal AOD750), as shown in Table 3.

### 3.4. Agreement of ACA Measurements between CASIA2 and Both Observers

There were no significant differences with the mean measurements derived from the CASIA2 automated software and observer 2 (range *p* = 0.05 to 0.99). However, nasal TISA500 was smaller for observer 1 compared to the CASIA2 with mean values of 0.161 ± 0.067 mm^2^ and 0.179 ± 0.060 mm^2^, respectively (*p* < 0.05). All the other parameters did not show any significant difference (range *p* = 0.06 to 0.95).

Reproducibility of CASIA2 measurements and those corresponding to observer 1 and 2 was excellent. CASIA2 measurements showed ICC values that ranged from 0.87 to 0.93 and from 0.90 to 0.96 compared to observer 1 and 2, respectively (Table 4). Bland–Altman figures illustrating the agreement between CASIA2 and both observers for the nasal and temporal ACA parameters can be found in Figure 3. A high level of agreement was observed with minimal variability among the measurements conducted by each of the observers and the CASIA2. The agreement between observer 1 and CASIA2 exhibited a slightly lower level of concordance compared to that observed between observer 2 and CASIA2.

## 4. Discussion

This study evaluated the ability of the built-in CASIA2 software to obtain ACA measurements and compared them with measurements obtained by ophthalmologists. The CASIA2 measurements not only demonstrated excellent repeatability for all parameters, but also when compared with the measurements obtained by the human graders, an excellent degree of agreement was also observed, except for nasal TISA500. To our knowledge, this is the first study evaluating the capability of the CASIA2 built-in software to obtain ACA measures derived from automatic SS location. Although previous studies suggested that SS placement should be conducted by a human grader, our study demonstrates that CASIA2 software can achieve consistent measurements in healthy subjects with high reliability.

The importance of automated SS detection lies in its potential to facilitate the evaluation and monitoring of patients with primary angle closure disease. In this study, we were able to demonstrate that the measurements derived from the automatic SS location of the CASIA2 were comparable to those of the human graders. However, nasal TISA500 measurements showed a significant difference compared to measurements obtained from the most experienced evaluator. In fact, this difference was also observed between the expert in AS-OCT analysis and the less experienced trained fellow. This might be explained by a small difference in the nasal SS location during the AS-OCT analysis. Previous studies have shown that a small change in the position of the SS mainly affects the parameters that measure size or angle configuration such as TISA500 and TISA750 [10]. However, even though there was a difference with the nasal TISA500 measurement in our study, the interobserver reproducibility and the CASIA2 vs. human graders reproducibility was excellent (ICC > 0.87). Interestingly, Tan et al. also evaluated the reproducibility of ACA measurements between expert and non-expert observers in AS-OCT analysis with the Visante OCT (Carl Zeiss, Meditec Inc., Dublin, CA, USA). They were able to demonstrate that the reproducibility of nasal and temporal ACA parameters between 2 experts and 23 non-experts was high. All the parameters showed excellent reproducibility with ICC means of 0.875, 0.942, and 0.906 for expert 1, expert 2, and non-expert observers, respectively. Nonetheless, the range of the coefficient of variation of ACA measurements was larger in the non-expert group [11].

In our study, we were also able to show that the reproducibility of the ACA measurements using the CASIA2 software tools to manually locate the SS was excellent for both evaluators (ICC > 0.96). In the past, to obtain the ACA measurements with some AS-OCT devices, it was necessary to export the images to an external software for the analysis. Nowadays, these machines have quite efficient software that allows an adequate location of the SS and the parameters derived from it. Xu et al. found that using the CASIA2 software to manually locate the SS, the reproducibility for AOD750, and TISA750 in healthy subjects was excellent with an ICC of 0.98 and 0.94, respectively [12]. Chan et al. evaluated the test–retest variabilities of different ACA parameters including AOD500 and TISA500 in healthy and primary angle closure subjects using two swept-source AS-OCTs, the CASIA2 and the ANTERION (Heidelberg Engineering, Heidelberg, Germany). They found that the CASIA2 repeatability coefficient for AOD500 and TISA500 were 0.035 mm and 0.014 mm^2^, respectively. Moreover, these repeatability coefficients measured with CASIA2 were found to be statistically significantly different, albeit slightly smaller, compared to those obtained from the ANTERION (*p* ≤ 0.001) [13]. In a multicenter study, Liu et al. evaluated the repeatability of ACA measurements in 116 healthy subjects with 171 eyes with primary angle closure disease. In their study, one single observer marked the nasal and temporal SS position of 18 evenly spaced B-scans over the 360° area of each eye. Repeatability coefficients for AOD750 and TISA750 were 0.058 mm and 0.030 mm^2^, respectively [14].

Recently, artificial intelligence has been used to obtain the location of the SS and subsequently be able to obtain the parameters derived from this. Xu et al. evaluated a deep neural network model for automatic SS detection using the CASIA SS-1000. These authors found that the distributions of prediction errors for the artificial intelligence model and intraobserver variability for the reference grader were similar. In fact, these prediction errors were significantly smaller than the interobserver variability seen between human expert graders [15]. Similarly, Pham et al. evaluated a deep convolutional neural network model for the location of the SS and found excellent agreement between the human grader and the artificial intelligence model. They also found that the ACA measurements repeatability for the artificial intelligence model was higher than the human graders [16]. Liu et al. used a deep learning model in the CASIA2 that allowed the automatic location of the SS and compared it with the manual location of an observer, finding that the deep learning model achieved a repeatability similar to that obtained by the human grader for TISA750 and AOD750 parameters. In fact, the authors report that the repeatability of these values was better with the deep learning model compared to the manual method with a higher mean absolute difference for the latter (0.0167 ± 0.0188 mm vs. 0.0210 ± 0.0212 mm, *p* = 0.003 for AOD750; 0.0093 ± 0.0093 mm^2^ vs. 0.0109 ± 0.0110 mm^2^, *p* = 0.018 for TISA750) [14]. One of the main advantages described for these artificial intelligence models is the ability to determine the location of the SS in less than 2 s, which is much faster than a human observer [14,16]. However, even though these models have shown a high repeatability in the measurement of the ACA parameters, they also have the limitation that they are trained on data that was originally labeled by human graders. This means that the models are likely to inherit any errors or biases that were present in the original data.

There is skepticism as to whether ACA measurements can be interchangeable between different platforms. Xu et al. evaluated the reproducibility and agreement of ACA measurements between the CASIA2 and Heidelberg Spectralis OCT2 AS-OCT (Heidelberg Engineering, Heidelberg, Germany). Their study showed good correlation for TISA750 and AOD750 (ICC = 0.78). However, there was poor agreement for anterior chamber (ICC = 0.20). These authors suggested that even though there was good agreement between both devices, the data from these two AS-OCTs should not be used interchangeably for clinical purposes [12]. Similarly, Chan et al. evaluated the agreement of ACA measurements between the CASIA2 and the ANTERION and found poor agreement for TISA500 and AOD500 (*p* < 0.001), suggesting that the measures of these devices may not be interchangeable [13]. Pardeshi et al. described that the differences between platforms could be related to poor image resolution or different wavelength when comparing the CASIA2 with the time domain and spectral domain AS-OCT. In their study, they evaluated the repeatability and agreement between two swept-source AS-OCTs, the CASIA SS-1000 and the ANTERION. They found that ACA measurements for both were nearly interchangeable [17,18]. However, differences between the CASIA2 and the ANTERION suggested that caution should be exercised when comparing ACA measurements across different OCT platforms, even if they share similar technology.

Our study has some limitations which should be considered. Firstly, the sample predominantly comprised individuals of Latin descent and only one type of AS-OCT scan. Therefore, caution should be exercised when extrapolating the results of our study to other populations or when using different types of AS-OCT scans. Secondly, our study focused solely on evaluating normal eyes, and it remains uncertain whether the reproducibility of our results could be replicated in eyes with glaucoma, primary angle closure, or other abnormalities affecting the angle geometry or structures. Finally, our study excluded individuals with a history of laser procedures, intraocular surgery, and use of topical medications that could modify the pupil size, among others. While these exclusion criteria were relevant for this specific study, they may limit the generalizability of the findings to broader populations. Therefore, further investigations are needed to assess the performance and reliability of the CASIA2 software in different populations and clinical scenarios.

In summary, this study demonstrates excellent repeatability in ACA measurements obtained through automatic SS detection using the CASIA2 built-in software. Furthermore, strong agreement was observed between these measurements and those provided by the two clinicians. However, despite the built-in software’s capability to accurately locate the SS and derive parameters from it, it is still suggested that human examiner supervision is needed to confirm the SS location. These findings highlight the potential value of parameters derived from the CASIA2 built-in software for clinical practice, particularly in assessing healthy individuals.

## Figures and Tables

**Figure 1 jcm-12-06381-f001:**
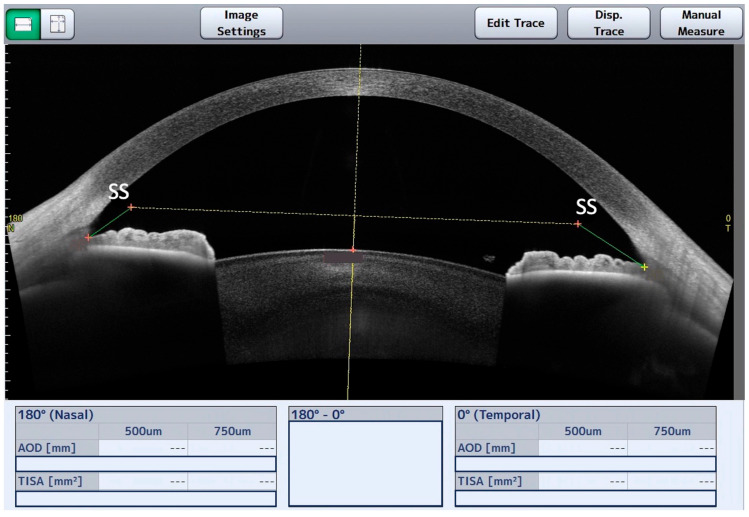
CASIA2 anterior segment OCT display after obtaining the automatic anterior chamber angle measurements and intentional relocation of the scleral spur (SS) that was performed by the masked observer. Nasal and temporal angle opening distance (AOD) and trabecular iris space area (TIA) are not available due to incorrect positioning of the SS.

**Figure 2 jcm-12-06381-f002:**
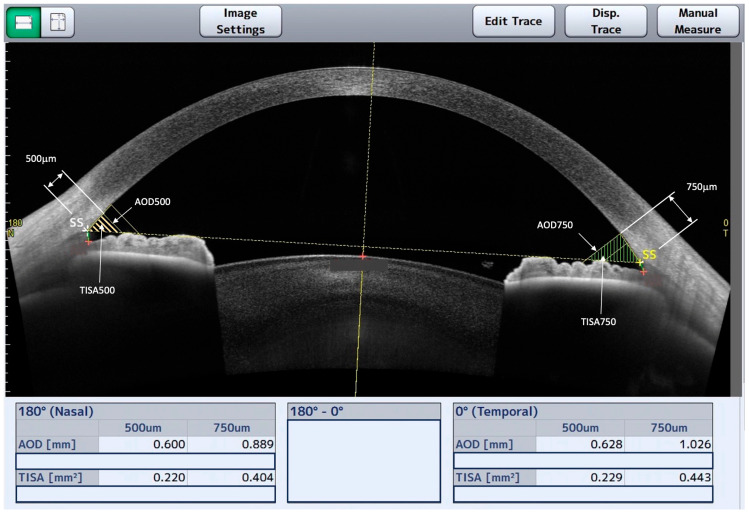
CASIA2 anterior segment OCT display after manual localization of the scleral spur (SS) was carried out by the human grader using the CASIA2 software tools to obtain the measurements of the anterior chamber angle parameters. Nasal and temporal angle opening distance (AOD) and trabecular iris space area (TISA) are shown in the image.

**Figure 3 jcm-12-06381-f003:**
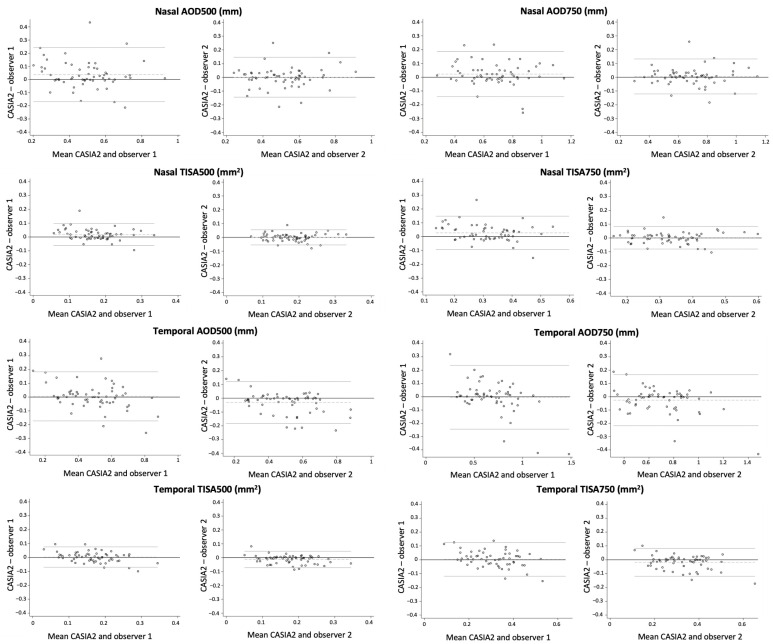
Bland–Altman plot: comparison of nasal and temporal ACA parameters between CASIA2 and both observers.

**Table 1 jcm-12-06381-t001:** CASIA2 automated software ACA parameters’ repeatability.

	Mean Capture 1	Mean Capture 2	*p*-Value	Mean Diff	ICC	ICC 95% CI
Temporal parameters						
AOD500 (mm)	0.490 ± 0.152	0.492 ± 0.152	0.94	−0.001	0.90	0.83–0.94
AOD750 (mm)	0.684 ± 0.212	0.690 ± 0.211	0.88	−0.005	0.88	0.80–0.93
TISA500 (mm^2^)	0.168 ± 0.057	0.166 ± 0.050	0.89	0.001	0.84	0.73–0.90
TISA750 (mm^2^)	0.316 ± 0.100	0.316 ± 0.093	0.99	0.000	0.87	0.78–0.92
Nasal parameters						
AOD500 (mm)	0.510 ± 0.156	0.497 ± 0.166	0.68	0.012	0.92	0.87–0.95
AOD750 (mm)	0.684 ± 0.186	0.679 ± 0.212	0.89	0.005	0.93	0.89–0.96
TISA500 (mm^2^)	0.180 ± 0.057	0.178 ± 0.060	0.88	0.001	0.91	0.85–0.95
TISA750 (mm^2^)	0.332 ± 0.097	0.328 ± 0.109	0.81	0.004	0.92	0.87–0.95

ICC = intraclass correlation coefficient; CI = confidence interval; ACA = anterior chamber angle; AOD = angle opening distance; TISA = trabecular iris space area.

**Table 2 jcm-12-06381-t002:** Intraobserver ACA measurements’ reproducibility.

	Mean Evaluation 1	Mean Evaluation 2	*p*-Value	Mean Diff	ICC	ICC 95% CI
Observer 1 Temporal parameters						
AOD500 (mm)	0.487 ± 0.190	0.478 ± 0.185	0.71	0.008	0.97	0.96–0.98
AOD750 (mm)	0.685 ± 0.267	0.671 ± 0.271	0.70	0.013	0.96	0.95–0.97
TISA500 (mm^2^)	0.166 ± 0.073	0.162 ± 0.071	0.67	0.004	0.96	0.95–0.98
TISA750 (mm^2^)	0.314 ± 0.129	0.308 ± 0.125	0.68	0.006	0.97	0.95–0.97
Nasal parameters						
AOD500 (mm)	0.469 ± 0.176	0.459 ± 0.171	0.66	0.009	0.97	0.96–0.98
AOD750 (mm)	0.656 ± 0.209	0.643 ± 0.206	0.61	0.013	0.97	0.96–0.98
TISA500 (mm^2^)	0.161 ± 0.067	0.157 ± 0.065	0.61	0.004	0.98	0.97–0.98
TISA750 (mm^2^)	0.303 ± 0.113	0.296 ± 0.110	0.61	0.007	0.98	0.97–0.98
Observer 2 Temporal parameters						
AOD500 (mm)	0.522 ± 0.180	0.537 ± 0.188	0.54	−0.014	0.98	0.97–0.98
AOD750 (mm)	0.713 ± 0.242	0.742 ± 0.261	0.38	−0.028	0.97	0.96–0.98
TISA500 (mm^2^)	0.182 ± 0.064	0.187 ± 0.068	0.51	−0.005	0.97	0.96–0.98
TISA750 (mm^2^)	0.337 ± 0.113	0.349 ± 0.123	0.42	−0.012	0.97	0.96–0.98
Nasal parameters						
AOD500 (mm)	0.505 ± 0.151	0.515 ± 0.160	0.61	−0.010	0.97	0.96–0.98
AOD750 (mm)	0.681 ± 0.189	0.698 ± 0.202	0.50	−0.017	0.98	0.97–0.99
TISA500 (mm^2^)	0.179 ± 0.057	0.183 ± 0.061	0.57	−0.004	0.97	0.96–0.98
TISA750 (mm^2^)	0.330 ± 0.097	0.338 ± 0.103	0.53	−0.008	0.98	0.97–0.98

ICC = intraclass correlation coefficient; CI = confidence interval; ACA = anterior chamber angle; AOD = angle opening distance; TISA = trabecular iris space area.

**Table 3 jcm-12-06381-t003:** Interobserver ACA measurements’ agreement.

	Mean Evaluation 1 Observer 1	Mean Evaluation 1 Observer 2	*p*-Value	Mean Diff	ICC	ICC 95% CI
Temporal parameters						
AOD500 (mm)	0.487 ± 0.190	0.522 ± 0.180	0.14	−0.035	0.95	0.93–0.96
AOD750 (mm)	0.685 ± 0.267	0.713 ± 0.242	0.40	−0.028	0.96	0.95–0.97
TISA500 (mm^2^)	0.166 ± 0.073	0.182 ± 0.064	0.08	−0.015	0.95	0.93–0.96
TISA750 (mm^2^)	0.314 ± 0.129	0.337 ± 0.113	0.16	−0.022	0.95	0.93–0.97
Nasal parameters						
AOD500 (mm)	0.469 ± 0.176	0.505 ± 0.151	0.09	−0.035	0.91	0.87–0.93
AOD750 (mm)	0.656 ± 0.209	0.681 ± 0.189	0.34	−0.024	0.94	0.92–0.96
TISA500 (mm^2^)	0.161 ± 0.067	0.179 ± 0.057	<0.05 *	−0.017	0.90	0.85–0.93
TISA750 (mm^2^)	0.303 ± 0.113	0.330 ± 0.097	0.06	−0.026	0.91	0.88–0.94

ICC = intraclass correlation coefficient; CI = confidence interval; ACA = anterior chamber angle; AOD = angle opening distance; TISA = trabecular iris space area. The asterisk (*) in the table indicates significant differences (*p* < 0.05).

**Table 4 jcm-12-06381-t004:** Agreement of ACA measurements between CASIA2 and both observers.

	Mean Observer Measurements	Mean CASIA2 Measurements	*p*-Value	Mean Diff	ICC	ICC 95% CI
Observer 1 Temporal parameters						
AOD500 (mm)	0.487 ± 0.190	0.491 ± 0.151	0.86	0.003	0.90	0.85–0.93
AOD750 (mm)	0.685 ± 0.267	0.687 ± 0.211	0.95	0.001	0.90	0.86–0.93
TISA500 (mm^2^)	0.166 ± 0.073	0.167 ± 0.054	0.08	0.001	0.88	0.82–0.91
TISA750 (mm^2^)	0.314 ± 0.129	0.316 ± 0.096	0.92	0.001	0.89	0.84–0.92
Nasal parameters						
AOD500 (mm)	0.469 ± 0.176	0.504 ± 0.160	0.11	0.034	0.88	0.81–0.92
AOD750 (mm)	0.656 ± 0.209	0.681 ± 0.199	0.35	0.025	0.93	0.90–0.95
TISA500 (mm^2^)	0.161 ± 0.067	0.179 ± 0.060	<0.05 *	0.017	0.87	0.77–0.92
TISA750 (mm^2^)	0.303 ± 0.113	0.330 ± 0.009	0.06	0.026	0.89	0.81–0.93
Observer 2 Temporal parameters						
AOD500 (mm)	0.522 ± 0.180	0.491 ± 0.151	0.15	−0.031	0.92	0.88–0.95
AOD750 (mm)	0.713 ± 0.242	0.687 ± 0.211	0.37	−0.026	0.93	0.90–0.95
TISA500 (mm^2^)	0.182 ± 0.064	0.167 ± 0.054	0.05	−0.014	0.90	0.83–0.94
TISA750 (mm^2^)	0.337 ± 0.113	0.316 ± 0.096	0.13	−0.021	0.92	0.87–0.95
Nasal parameters						
AOD500 (mm)	0.505 ± 0.151	0.504 ± 0.160	0.96	−0.001	0.93	0.91–0.95
AOD750 (mm)	0.681 ± 0.189	0.681 ± 0.199	0.98	0.000	0.96	0.94–0.97
TISA500 (mm^2^)	0.179 ± 0.057	0.179 ± 0.060	0.99	0.000	0.92	0.89–0.94
TISA750 (mm^2^)	0.330 ± 0.097	0.330 ± 0.009	0.96	0.000	0.94	0.91–0.95

ICC = intraclass correlation coefficient; CI = confidence interval; ACA = anterior chamber angle; AOD = angle opening distance; TISA = trabecular iris space area. The asterisk (*) in the table indicates significant differences (*p* < 0.05).

## Data Availability

All the obtained data used to support the findings of this study are available from the corresponding author upon reasonable request.

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
