# Peer review of "Agreement and Reproducibility of Anterior Chamber Angle Measurements between CASIA2 Built-In Software and Human Graders"

_jcm, 2023, doi:10.3390/jcm12196381_

Round 1

Reviewer 1 Report

Thank you for the opportunity to review this paper. It is a very interesting job about the possibility to use a standardized tool to evaluate anterior chamber. However, I have some questions:

1. Lines 112-114: "The upper eyelid was gently raised and a gentle traction of the lower eyelid was made by the operator taking care to avoid inadvertent pressure on the eyeball". Has been this action performed for all the patients? and why? It seems that the instrument can not acquire a good image without operator's help. Is it really so? In this case, it could be a potential bias in the clinical practice. Please clarify.

2. Lines 114-115: " The image across the horizontal meridian was used for the analysis". What about vertical meridian? It is well known that there is an important difference in widening between superior and inferior anterior chamber angle. Please clarify.

Author Response

I would like to express my sincere gratitude to the reviewer for dedicating their time and expertise to review my article. Your valuable insights and feedback are greatly appreciated. In the following responses, I will address each of your comments and suggestions in detail. 

3. Point-by-point response to Comments and Suggestions for Authors

Comments 1: - Lines 112-114: "The upper eyelid was gently raised and a gentle traction of the lower eyelid was made by the operator taking care to avoid inadvertent pressure on the eyeball". Has been this action performed for all the patients? and why? It seems that the instrument can not acquire a good image without operator's help. Is it really so? In this case, it could be a potential bias in the clinical practice. Please clarify

Response 1: We performed this action consistently for all patients to ensure uniformity in our data collection protocol. In clinical practice, we have observed that while horizontal meridian captures tend to be easier to obtain and yield high-quality images, a small subset of subjects may experience slight peripheral coverage by the eyelid, which could potentially affect subsequent image analysis. Therefore, we opted for a standardized approach for all patients. It's important to note that the eyelid manipulation was done gently and minimally, and we do not believe it introduced any significant interference with the scan's reliability.

Comments 2: - Lines 114-115: " The image across the horizontal meridian was used for the analysis". What about vertical meridian? It is well known that there is an important difference in widening between superior and inferior anterior chamber angle. Please clarify.

Response 2: We acknowledge that the difference in widening between the superior and inferior anterior chamber angle is a well-documented phenomenon. However, we regret to inform you that we did not include the vertical meridian in our study. While we recognize that this aspect of the angle's behavior is of clinical significance, our study primarily focused on the horizontal meridian.

Reviewer 2 Report

Overall, the paper is well-written and addresses the question of whether the automated software in-built into the AS-OCT device can be used for angle measurements. In particular, the discussion section is well-written and I enjoyed reading it.

Line 53 and 54: I was curious to know where in the world gonioscopy was not performed even for glaucoma patients. I would suggest to include the location instead of generalizing it. Perhaps, you could say something like “as per the reports/ study conducted in (country), …..”

Line 82 through 97: Any reason why the participants were screened so strictly for glaucoma including RNFL and visual fields? Does a study comparing 2 methods of a measurement require such a strict criteria of participant selection? Was this study part of a larger study, in which case it needs to be mentioned? 

Line 101: data loss possibly due to? Please mention the possible reasons for anticipated data loss.

Line 120: I think it is important to give the definitions of parameters TISA and AOD in this paper instead of referring the readers to another article to understand their definition, especially when the entire paper is comparing these parameters between the 2 methods.

Line 121, 122: It would be useful to the readers if there was an image of AS-OCT depicting the location of SS, the portion of the image considered as TISA and AOD.

Figures S1 and S2: There are many abbreviations in the figure that the readers may not be familiar with. It will be useful to expand at least the relevant ones. For example, right on the image, there is a point called “AR”, which is not expanded.

Line 132: Were the images shuffled or counter-balanced in the second analysis?

Line 184 and 198: Mention that this table is for capture 1 or capture 2 or the average of the two captures.

Line 214: You can mention about the variability in the agreement in some of the parameters between the observers and the device.

Line 318: Why do you think that the sample size small although you calculated the required number of participants for the desired outcome?

Good. There are a few spelling errors.

Author Response

I would like to express my sincere gratitude to the reviewer for dedicating their time and expertise to review my article. Your valuable insights and feedback are greatly appreciated. In the following responses, I will address each of your comments and suggestions in detail.Each comment that had modifications in the main document will be highlighted in a commentary in the .docx file.

Point-by-point response to Comments and Suggestions for Authors

Comments 1: - Line 53 and 54: I was curious to know where in the world gonioscopy was not performed even for glaucoma patients. I would suggest to include the location instead of generalizing it. Perhaps, you could say something like “as per the reports/ study conducted in (country), …..”

Response 1: Thank you for pointing this out. We agree with this comment. Therefore, we have updated this paragragh. Previous studies conducted in the United States, specifically evaluating Medicare claims, have shown that less than half of glaucoma patients underwent gonioscopy even before glaucoma procedures. This finding is intriguing and may potentially reflect practice patterns in other regions.

Comments 2: - Line 82 through 97: Any reason why the participants were screened so strictly for glaucoma including RNFL and visual fields? Does a study comparing 2 methods of a measurement require such a strict criteria of participant selection? Was this study part of a larger study, in which case it needs to be mentioned? 

Response 2: We implemented stringent participant screening criteria, including RNFL and visual field assessments, to minimize any potential bias associated with the inclusion of individuals with glaucoma or glaucoma suspects. Our aim was to align with the methodological standards established in previous studies focusing on repeatability and reproducibility. Regarding your inquiry about the study's context, this research was conducted as an independent study, and we appreciate your suggestion to clarify this point.

Comments 3: - Line 101: data loss possibly due to? Please mention the possible reasons for anticipated data loss.

Response 3: We appreciate your valuable feedback. In our study, despite being cross-sectional, data collection occurred on different days from the clinical evaluation. Given the challenges of maintaining strict patient adherence to research protocols in our specific environment, we had to adjust the sample size accordingly

Comments 4: - Line 120: I think it is important to give the definitions of parameters TISA and AOD in this paper instead of referring the readers to another article to understand their definition, especially when the entire paper is comparing these parameters between the 2 methods

Response 4: We agree that it is essential to provide clear definitions of the parameters TISA and AOD within the manuscript, rather than referring readers to another article for their definitions. We will make the adjustments to ensure that these definitions are included in our paper (line 121-127). Your suggestion aligns with our goal of making the paper self-contained and accessible to a broader audience.

Comments 5: - Line 121, 122: It would be useful to the readers if there was an image of AS-OCT depicting the location of SS, the portion of the image considered as TISA and AOD

Response 5: We appreciate your feedback and agree that including an image of AS-OCT depicting the location of SS, the portion of the image considered as TISA, and AOD would be beneficial for our readers. We will make some modifications to supplementary figures and add them as actual figures of the main document.

Comments 6: - Figures S1 and S2: There are many abbreviations in the figure that the readers may not be familiar with. It will be useful to expand at least the relevant ones. For example, right on the image, there is a point called “AR”, which is not expanded

Response 6: We acknowledge the concern regarding the use of abbreviations in Figures S1 and S2. We will make the necessary adjustments to expand and clarify the relevant abbreviations to enhance reader comprehension. Your input is greatly appreciated in improving the clarity of our manuscript.

Comments 7: - Line 132: Were the images shuffled or counter-balanced in the second analysis?

Response 7: Shuffled. Each capture was identified with a number and assigned in a shuffle order with respect to the first analysis.

Comments 8: - Line 184 and 198: Mention that this table is for capture 1 or capture 2 or the average of the two captures.

Response 8: The table corresponds to the average of the two captures, as stated in the Methods section between lines 152 and 155 (of the updated version).

Comments 9: - Line 214: You can mention about the variability in the agreement in some of the parameters between the observers and the device

Response 9: We agree. We will add comments about the variability in agreement (line 230-236)

Comments 10: - Line 318: Why do you think that the sample size small although you calculated the required number of participants for the desired outcome?

Response 10: You're absolutely right, and I appreciate your observation. The study did indeed have the required statistical power to achieve its objectives. We deleted the observation.
